# Chinese-style incentives: The intraindustry ripple effects of CEO awards

Yu Wu●*, Yingyi Hu

Institute of Chinese Financial Studies, Southwestern University of Finance and Economics, Wenjiang District, Chengdu, China

* 117020204016@samil.swufe.edu.cn

## Abstract

Considering the background of traditional Chinese culture, which emphasizes that "when we see outstanding people, we should think of emulating them", and social comparison theory, this study explores how CEO awards impact the R&D investment of award-winning CEOs' competitors. The results show that award-winning CEOs' competitors increase R&D investment in the postaward period relative to the preaward period. We further find that CEO awards' "gold content", the social attention of award-winning CEOs' competitors, the similarity between award-winning CEOs and their competitors, and industry competitive pressure are important factors affecting the size of ripple effects. Empirical evidence also shows that the intraindustry ripple effects of CEO awards significantly improve the firm performance and value of competitors. In a robustness test, we confirm CEO awards' intraindustry ripple effects from the perspective of the number of patent applications. The ripple effects of CEO awards are still valid after using PSM-DID to alleviate endogeneity problems and considering the right-side distribution of R&D investment.

## Introduction

As nonfinancial incentives, CEO awards serve not only to recognize entrepreneurs' past achievements but also provide incentives to potential competitors of award-winning CEOs because the social status and recognition of CEOs will increase significantly after winning awards. Especially in China, where traditional culture emphasizes that when "we see outstanding people, we should think of emulating them". Maslow's hierarchy of needs divides human needs from low to high levels, including physiological needs, security needs, social needs, respect needs, and self-realization needs. CEO pay and job stability generally belong to security needs and social needs, and respect and social influence belong to the respect and self-realization needs. When low-level demands are satisfied, their incentive effect will be reduced, while high-level demands can have a stronger incentive effect. As nonfinancial incentives, CEO awards can lead to salary and job stability improve and meet CEOs' low-level needs, as well as allowing them to achieve respect, self-fulfillment and other higher-level needs. Social comparison theory emphasizes that CEOs are driven by social status and recognition and have a strong motivation for upward comparison, which means they have strong incentives to compare themselves with award-winning CEOs. Therefore, given China's cultural background combined alongside the insights of social comparison theory, taking award-winning CEOs as the

**Data Availability Statement:** All relevant data are within the paper and its Supporting Information files.

**Funding:** The authors received no specific funding for this work.

**Competing interests:** The authors have declared that no competing interests exist.

reference points for upward comparison significantly impacts competitors' investment decisions. Existing studies mostly discuss the potential impact of CEO awards from the perspective of CEO compensation [1–3], job stability [4], risk-taking [5], corporate performance [6, 7], earnings management [8], and corporate strategy [9, 10]. Few studies analyze the ripple effects of CEO awards. The study of the intraindustry ripple effects of CEO awards not only helps us to further understand the incentive effects of nonmaterial rewards but also has important theoretical and practical significance for the construction of managers' incentive philosophy, in line with the actual Chinese situation and cultural traditions.

As a common psychological phenomenon, social comparison can help individuals understand themselves more clearly, discover their strengths, or identify their shortcomings, which affect the actions taken by individuals in the next step [11]. On this basis, Smith [12] further divided social comparison into upward comparison and downward comparison. Upward comparison refers mainly to self-evaluation by comparing oneself to outstanding individuals and finding methods to improve oneself [13–15]. Park and Westphal [16] find that CEOs who are driven by social recognition and status are more likely to achieve self-evaluation through upward social comparison. Therefore, upward social comparison can provide positive incentives for CEOs and encourage them to take actions to catch up with "successful" CEOs to obtain social status and recognition [17]. Innovation is not only the core driving force of macroeconomic growth but also an important way for enterprises to form long-term competitive advantages and improve corporate performance. Therefore, award-winning CEOs provide incentives to competitors to increase R&D investment to enhance corporate performance and corporate value. Compared with the existing conclusion that CEO awards cause competitors to undertake more inefficient acquisition activities, we define the ripple effects of CEO awards in China as a kind of Chinese-style incentive that is tightly related to China's traditional culture.

The main contributions and implications of this paper are as follows.

(1) The existing literature focuses mainly on the impact of CEO awards on the enterprise, including the individual effect and organizational effect, and pays less attention to the intraindustry ripple effects of CEO awards. Based on social comparison theory and motivation theory, this study proposes that CEOs invest more in R&D after witnessing other CEOs winning awards; this is the intraindustry ripple effect of CEO awards. We also discuss the influencing factors of the ripple effects of CEO awards under the Chinese traditional cultural background. The conclusions of this study enrich the relevant research on CEO characteristics and provide a novel perspective for a comprehensive understanding of the incentive effects of nonfinancial rewards, such as CEO awards.

(2) According to Maslow's hierarchy of needs, CEOs' need for respect and self-realization becomes more prominent after their basic physiological and security needs are met. Existing research focuses mainly on the incentive effects of salary arrangements on CEOs, ignoring the incentive effect of satisfying the higher-level needs of CEOs. CEO awards, as nonfinancial incentives, not only impact the internal decisions of firms, such as risk taking [5] or business strategy [9, 10], but also have ripple effects on award-winning CEOs' competitors, which stimulate them to invest more in R&D to increase firm performance and value.

## Literature review and research hypotheses

### CEO awards: Positive individual effects and negative organizational effects

In terms of individual effects, CEOs' salaries increase significantly after winning awards. Especially in firms with strong performance, awards help CEOs obtain compensation that is greater

than the industrial average [1, 2]. Cu Weihua and Li Peigong [3] found that media attention strengthened award-winning CEOs' influence, and CEOs were rewarded with substantial pay. Moreover, not only do CEOs obtain higher salaries as a result of winning awards, but their work partners can also obtain higher salaries and become more likely to be promoted to a CEO position through the internal and external manager markets [18]. In addition, awards enhance the stability of CEOs' position, with other executives within an award-winning CEO's team more likely to be scapegoats for poor firm performance, which means that they are more likely to be dismissed because of poor performance [4].

From the perspective of organizational effects, CEOs winning awards even negatively impacts firm performance. These studies found that enterprise performance sustains a decline after CEOs win awards [1, 2]. In particular, superstar CEOs' self-identity leads to strategic rigidity when the uncertainty of the business environment rises, seriously damaging firm performance [7]. Existing studies also point out that whether CEOs can win awards depends mainly on historical performance, resulting in the decline of corporate performance after CEOs win awards [6]. In addition, CEOs tend to issue positive unexpected earnings or even utilize earnings management to whitewash performance to maintain a superstar identity [2, 8]. From the perspective of corporate strategy, it has been found that the adjustment of successful strategies is regarded as a high-risk behavior for award-winning CEOs. CEOs tend to stick to the corporate strategy that allows them to win and are unwilling to adjust to environmental changes. Superstar CEOs even stick to M&A strategies that have proven unsuccessful [9, 10].

## Intraindustry ripple effects of CEO awards

Festinger [19] proposed social comparison theory, which emphasizes that people tend to use similar persons as reference points in self-evaluation [20]. Social comparison can help individuals clearly understand themselves, including discovering their strengths or determining their weaknesses, which affects the direction of individual action in the next step [11]. Ager et al. [21] analyzed more than 5,000 German pilots in World War II and found that the notification of rewards to a single pilot would stimulate other members in the same flight squadron, increasing the mortality rate. Shi et al. [22] found that award-winning CEOs significantly stimulated competitor CEOs to undertake more acquisition activities to increase their social status and recognition; this phenomenon is the ripple effects of CEO awards. Deng Bofu et al. [23] found that corporate awards in China increase the pressure on competitor CEOs, which causes significant earnings management.

Park and Westphal [16] pointed out that CEOs are likely to perform self-evaluations through upward social comparison. That is, upward social comparison can provide positive incentives for CEOs and encourage them to take actions to catch up with "successful" persons to obtain social status and recognition [17]. Chinese traditional culture also emphasizes peer pressure generated by comparisons with outstanding persons, thus generating a self-motivation effect. CEO awards allow the media to attribute strong firm performance to the CEO [9]. For example, in Forbes China's awards, Forbes considers the market value of listed companies, stock price fluctuations, the net profit and growth rate of operating income, ROA, ROE and other basic financial indicators and requires the CEO to work in the same firm for the previous three years. After the CEO wins an award, the outside world is more likely to evaluate him/her as the industry benchmark, and the CEO obtains higher social status and recognition, that is, he/she becomes the upward comparison reference point. Therefore, award-winning CEOs may stimulate their competitors to increase R&D investment to enhance firm performance and value.

The consideration of award-winning CEOs as a benchmark for upward comparison, the external media attribute firm performance to award-winning CEOs, as well as Chinese

society's emphasis on cultural background strengthen the performance demands of non-award-winning CEOs, who have the motivation to take action to improve their firms' long-term performance and value. Innovation is not only the core driving force of macroeconomic growth but also an important factor for firms to form long-term competitive advantages and improve firm performance. In a traditional cultural context, strong upward comparison motivation magnifies the long-term performance demands of other CEOs when intraindustry CEOs win awards, and R&D investment provides an effective way for CEOs to obtain long-term competitive advantages and performance growth. Thus, this paper puts forward the following hypothesis:

**H1**: **CEO awards have significant intraindustry ripple effects, which provide incentives for award-winning CEOs' competitors to enhance R&D investment.**

Award-winning CEOs' selection by the media as the reference points for upward comparison may have ripple effects on other CEOs. Specifically, the ripple effects of CEO awards are influenced by the following factors.

First, CEO awards can bring additional private benefits [24–26]. The existing literature has confirmed that CEO awards bring both financial and nonfinancial rewards, such as salary increases, position stability, higher social status and a stronger reputation [1–3]. Specifically, the salary level of CEOs after winning awards has increased significantly, even their work partners [1–3]. Other members of the team are likely to become "scapegoats" for award-winning CEOs, which means that the job stability of award-winning CEOs is higher than that of non-award-winning CEOs [4]. In particular, media reports on CEOs can significantly improve their salary levels if award-winning CEOs become superstars in the industry [3]. Therefore, the ripple effects are affected by the degree of social recognition of CEO awards, that is, the "gold content" of the awards. On the other hand, if the competitors of award-winning CEOs already have a high degree of social attention, it is not necessary to obtain long-term performance through R&D innovation, which will reduce the ripple effects of CEO awards. Accordingly, this paper proposes Hypothesis H2:

**H2**: **The intraindustry ripple effects are affected by the gold content of CEO awards and the social attention of award-winning CEOs' competitors.**

Second, the reference point is the similarity of award-winning CEOs to competitors involved in social comparison. In general, CEOs in the same industry face similar business environments, which ensure that upward social comparison can produce incentive effects [27]. Therefore, the size of the ripple effects is affected by the similarity between award-winning CEOs and competitors. CEO age and the property rights nature of the enterprise can be used to measure whether CEOs belong to the same group [22]. First, CEOs of similar ages are more likely to establish connections with people with similar professional and educational experiences, and people of similar ages often have similar values. In addition, Chinese society is influenced by Confucian culture; thus, if the age of award-winning CEOs is less than the age of competitors, the motivation for competitors to compare themselves with award-winning CEOs is weakened. Second, there are differences between state-owned enterprises and non-state-owned enterprises in terms of management mode and business objectives, and state-owned enterprises have a low willingness to compare themselves with non-state-owned enterprises. Accordingly, this paper proposes the following hypothesis:

**H3**: **The ripple effects of CEO awards are affected by the similarity between upward comparison groups.**

The high similarity of products among enterprises means that they belong to the same strategic group in the industry [28], which increases the social comparison between CEOs [29, 30]. Kim and Tsai [31] found that CEOs with competitive relationships are more likely to compare themselves with each other, which affects the decision-making of enterprises to develop new markets [32], adopt new technologies [33], or reform organizational management. Existing studies also point out that CEOs with competitive relationships are more likely to adjust their investment strategies with potential competitors [34]. That is, the degree of industry competition may amplify the ripple effects of CEO awards. Accordingly, this paper proposes the following hypothesis:

**H4**: **The ripple effects of CEO awards are affected by the degree of industry competition.**

## Data description and research design

### Data on CEO awards

The original data for this paper are from the CSMAR database. The initial sample included all A-share listed companies from 2007 to 2018 and was screened according to the following procedures: (1) financial listed companies were deleted; (2) companies listed for less than three years were deleted; (3) firms in industries with fewer than 5 listed companies were deleted; and (4) all continuous variables were treated with 1% and 99% quantile tail reductions. The final sample includes 25585 observations of approximately 2573 listed companies. This paper aims to examine the impact of CEO awards on R&D investment. The primary task is to define the scope of CEO awards by referring to the existing literature [3, 5, 6], including "China 's annual economic figures" selected by CCTV, "Best CEO of Listed Companies" selected by Forbes China, "Best Business Leader of China" selected by First Finance, "China 's most influential business leaders" selected by Wealth, and "Top Ten Outstanding CEOs of China" selected by the China Times. Table 1 shows the statistics of CEO awards from 2007 to 2018. Overall, CEO awards are highly rare.

### Variable definitions

R&D investment. This paper discusses corporate innovation behavior from the perspective of R&D investment: R&D investment 1 (*R&D_*1) is corporate R&D expenditure divided by total

**Table 1. Distribution of CEO awards.**

| Year | Award-winning CEOs | Listed companies | Percentage |
|---|---|---|---|
| 2007 | 28 | 1354 | 2.0679% |
| 2008 | 25 | 1479 | 1.6903% |
| 2009 | 55 | 1518 | 3.6232% |
| 2010 | 49 | 1676 | 2.9236% |
| 2011 | 25 | 2032 | 1.2303% |
| 2012 | 21 | 2263 | 0.9280% |
| 2013 | 30 | 2349 | 1.2771% |
| 2014 | 34 | 2408 | 1.4120% |
| 2015 | 49 | 2507 | 1.9545% |
| 2016 | 60 | 2644 | 2.2693% |
| 2017 | 41 | 2652 | 1.5460% |
| 2018 | 33 | 2703 | 1.2209% |
| Total | 450 | 25585 | 1.7588% |

Note: Prior to 2012, rankings included only Chinese A-share listed companies. Beginning in 2012, Forbes' Chinese CEO ranking samples included Chinese companies listed in the A-share, Hong Kong share and US share markets.

assets at the end of the period; R&D investment 2 (*R&D_2*) is the division of enterprise R&D expenditure by current operating income.

**CEO awards.**   When there was a CEO award in the same industry in the previous year, *Ind* equals 1 in that year, and 0 otherwise. When there was a CEO award in the same industry and area in the previous year, *Ind_Area* equals 1 in that year, and 0 otherwise.

**Control variables.**   This paper controls for the company's operating conditions, corporate governance, financial conditions and other factors: the company's operating conditions include the firm's size, asset-liability ratio, company growth capacity, profitability, and years of establishment; corporate governance factors include ownership concentration and property rights nature; financial conditions include the proportion of cash assets and operating cash flow per share. This paper also controls the regional marketization index to reduce the impact of regional factors on the results. Symbols and definitions for the variables are shown in Table 2.

## Descriptive statistics

Table 3 reports the statistical descriptive results for the main variables in this paper. According to the industry distribution of CEO awards, more than 52% of enterprises have ever had award-winning CEOs, which indicate that the data adopted in this paper have good industry coverage. Furthermore, the proportion of CEO awards in the same industry and region in the sample exceeds 11%, which is smaller than the proportion of CEO awards in the same industry, indicating that CEO awards have a certain degree of regional agglomeration. The mean values of R&D investment are 1.3170 and 2.5450, and the standard deviations are 1.6957 and 3.3768, respectively, which indicate that there are large differences in R&D investment among different enterprises in the sample. The control variables are not significantly different from those in the existing research.

## Empirical analysis

### Firm performance and the probability of CEOs winning awards

According to the above analysis, whether CEOs win awards is tightly related to firm performance. CEOs with better firm performance are more likely to gain recognition within and

**Table 2.  Variable definitions.**

| Types | Symbol | Definition and Calculation |
|---|---|---|
| Dependent variables | *R&D_1* | R&D investment/Total assets (%) |
| | *R&D_2* | R&D investment/Operating receipt (%) |
| Independent variables | *Ind* | Dummy variable, the detailed calculation method is shown above |
| | *Ind_Area* | Dummy variable, the detailed calculation method is shown above |
| Control Variables | *size* | Ln (Total asset) |
| | *lev* | Total debt/Total asset |
| | *roa* | Net profit/Total asset |
| | *growth* | Growth rate of operating receipt |
| | *cash* | Cash/Total assets |
| | *fcf* | Net cash flow/Total capital stock |
| | *top5* | Shareholding of top 5 shareholders (%) |
| | *age* | Ln (year of establishment) |
| | *soe* | Dummy variable equal to 1 if the firm is state-owned, 0 otherwise |
| | *MRK* | Regional marketization index |

**Table 3. Description of the data.**

| Variable | N | Mean | Media | Standard error | Min | Max |
|---|---|---|---|---|---|---|
| R&D_1 | 25585 | 1.3170 | 0.6916 | 1.6957 | 0 | 9.1325 |
| R&D_2 | 25585 | 2.5446 | 1.1968 | 3.3768 | 0 | 15.4099 |
| Ind | 25585 | 0.5231 | 0 | 0.4994 | 0 | 1 |
| Ind_Area | 25585 | 0.1147 | 0 | 0.3186 | 0 | 1 |
| size | 25585 | 20.1025 | 20.0648 | 1.7239 | 14.8652 | 24.6293 |
| lev | 25585 | 0.4603 | 0.4547 | 0.2248 | 0.0515 | 1.1479 |
| roa | 25585 | 0.0384 | 0.0367 | 0.0609 | -0.2434 | 0.2133 |
| growth | 25585 | 0.4289 | 0.1195 | 1.1935 | -0.7522 | 7.4735 |
| cash | 25585 | 0.1652 | 0.0971 | 0.1898 | 0.0013 | 0.8005 |
| fcf | 25585 | 0.3586 | 0.2694 | 0.7865 | -2.2922 | 3.4016 |
| top5 | 25585 | 52.8262 | 53.1111 | 15.6882 | 18.3700 | 87.9449 |
| age | 25585 | 8.6185 | 8.6769 | 0.3979 | 6.0450 | 10.6797 |
| soe | 25585 | 0.2164 | 0 | 0.4118 | 0 | 1 |
| MRK | 25585 | 7.7941 | 7.9100 | 1.8834 | -0.3000 | 11.1092 |

outside the industry. The implicit assumption is that the probability of CEOs winning awards is significantly related to firm performance. Under this condition, other CEOs in the industry will have the motivation to improve their long-term performance by increasing R&D investment in the process of upward comparison. This paper sets the following probit regression model:

$$\mathrm{Pr}obit(WIN_{i,t}) = \alpha_0 + \alpha_1 roa_{i,t} + \alpha_2 roa[-1]_{i,t} + \alpha_3 roa[-2]_{i,t} + \sum_j \alpha_j X_{j,i,t} + \sum ind + \sum year + \varepsilon_{i,t} \quad (1)$$

where CEO awards (*WIN*) is a dummy variable that equals 1 if at least one CEO of the firm has won an award and 0 otherwise. *roa* is the current rate of return on the total assets of the enterprise, *roa*[−1] is the rate of return on total assets lagged one period, *roa*[−2] is the rate of return on total assets lagged two periods, and *X* is a control variable for industry and year effects. Table 4 empirically examines the relationship between firm performance and the probability of CEOs winning awards.

The results in Table 4 show that the current return on total assets, the one-period lagged return on total assets, and the two-period lagged return on total assets are all significantly positively correlated with the probability of CEOs winning, which indicates that the probability of CEOs winning is greater when the long-term performance of enterprises rises steadily. The results of this paper further confirm the conclusions of Chen Hong et al [6]: CEOs with better historical performance have a higher probability of winning awards.

## CEO awards and R&D investment by competitors

To identify the relation between CEO awards and R&D investment by competitors, we set the following regression:

$$R\&D_{i,t} = \alpha_0 + \alpha_1 Ind_{i,t}/Ind\_Area_{i,t} + \sum_j \alpha_j X_{j,i,t} + \sum ind + \sum year + \varepsilon_{i,t} \quad (2)$$

The explained variables include R&D investment and control variables, as shown in Table 2. The industry effect and year effect are controlled in the regression model. If CEO awards have positive intraindustry ripple effects, they should be significantly positively related to R&D investment; that is, CEO awards can drive competitors to undertake more R&D

**Table 4. Firm performance and the probability of CEOs winning awards.**

| Variable | Probability of the CEO winning an award | |
|---|---|---|
| | **(1)** | **(2)** |
| roa | 4.8226*** (0.4602) | 5.8000*** (0.5589) |
| roa[−1] | 3.9551*** (0.4353) | 4.5220*** (0.5086) |
| roa[−2] | 0.9298*** (0.2364) | 1.3840*** (0.2570) |
| size | | 0.1730*** (0.0175) |
| lev | | 0.9152*** (0.0139) |
| growth | | 0.0357* (0.0196) |
| cash | | -0.4677*** (0.1555) |
| fcf | | 0.0476*** (0.0276) |
| top5 | | -0.0580 (0.0156) |
| age | | -0.0356 (0.0693) |
| soe | | -0.6414*** (0.0778) |
| MRK | | 0.1071*** (0.0161) |
| Industry | Y | Y |
| Year | Y | Y |
| N | 25585 | 25585 |
| Pseudo $R^2$ | 0.1456 | 0.2284 |

investment. The regression results of model (2) shown in Table 5. In columns (1) and (2), the regression coefficients of *Ind* and *Ind_Area* are 0.3066 and 0.3827, respectively, and significant at the 1% level. The results show that CEO awards have intraindustry ripple effects, which can promote the level of R&D investment of other enterprises. Moreover, the coefficient of *Ind_Area* is larger than that of *Ind*, which indicates that the ripple effects are more obvious

**Table 5. CEO awards and R&D investment by competitors.**

| Variables | R&D_1 | | R&D_2 | |
|---|---|---|---|---|
| | **(1)** | **(2)** | **(3)** | **(4)** |
| Ind | 0.3066*** (0.0184) | | 0.6755*** (0.0360) | |
| Ind_Area | | 0.3827*** (0.0290) | | 0.7949*** (0.0567) |
| size | -0.0349*** (0.0064) | -0.0386*** (0.0064) | -0.1453*** (0.0125) | -0.1536*** (0.0125) |
| lev | -0.4755*** (0.0472) | -0.4827*** (0.0473) | -2.5122*** (0.0922) | -2.5284*** (0.0925) |
| roa | 2.5229*** (0.1645) | 2.5528*** (0.1649) | 1.1218*** (0.3211) | 1.1967*** (0.3222) |
| growth | 0.0132* (0.0079) | 0.0136* (0.0079) | 0.0835*** (0.0154) | 0.0844*** (0.0154) |
| cash | 0.4245*** (0.0507) | 0.4316*** (0.0508) | 0.7712*** (0.0990) | 0.7868*** (0.0994) |
| fcf | 0.0427*** (0.0121) | 0.0424*** (0.0121) | -0.0566*** (0.0236) | -0.0572** (0.0236) |
| top5 | -0.0016** (0.0006) | -0.0014** (0.0006) | -0.0042*** (0.0011) | -0.0040*** (0.0012) |
| age | -0.4514*** (0.0262) | -0.4452*** (0.0262) | -1.1768*** (0.0511) | -1.1644*** (0.0513) |
| soe | 0.0679*** (0.0238) | 0.0728*** (0.0239) | 0.0837* (0.0464) | 0.0942*** (0.0466) |
| MRK | 0.1603*** (0.0052) | 0.0939*** (0.0053) | 0.1506*** (0.0101) | 0.1251*** (0.0104) |
| Industry effect | Y | Y | Y | Y |
| Time effect | Y | Y | Y | Y |
| N | 22932 | 22932 | 22932 | 22932 |
| Adj.$R^2$ | 0.3676 | 0.3648 | 0.4172 | 0.4132 |

Note: In this table, we use only those firms in which CEOs do not win awards as a subsample.

***, **, and * indicate significance at 1%, 5% and 10%, respectively.

when award-winning CEOs and competitors have the same industry and region. In columns (3) and (4), the proportion of corporate R&D investment in operating income is used as the explained variable, and the results still support the above conclusions.

## The influencing factors of CEO award ripple effects

### The impact of CEO awards' gold content and competitors' social attention

In 2012, Forbes China extended the best CEO selection range to Chinese firms listed in Hong Kong and the United States. The expansion of the selection range meant that A-share listed companies would compete with well-known companies, such as Tencent, Alibaba, Jingdong, and Xiaomi, thereby increasing the "gold content" of the awards. The awarded CEOs receive more social attention, which may amplify the intraindustry ripple effects of CEO awards. To examine the impact of the gold content of CEO awards, we set a dummy variable, $IV\_2012$, equal to 0 for the period before 2012 and 1 for the period after 2012. If the regression coefficient of $Ind\_Area^*IV\_2012$ is significantly positive, it shows that the increase in gold content of CEO awards after 2012 has amplified the ripple effects. At the same time, we use the number of media reports on CEOs to represent external attention. If the regression coefficient of $Ind\_Area^*media$ is significantly negative, it indicates that CEOs' higher social attention weakens the ripple effects of CEO awards.

$$R\&D_{i,t} = \alpha_0 + \alpha_1 Ind\_Area_{i,t} + \alpha_2 Ind\_Area_{i,t} * IV\_2012_{i,t}/media_{i,t}$$
$$+ \sum_j \alpha_j X_{j,i,t} + \sum ind + \sum year + \varepsilon_{i,t} \tag{3}$$

Table 6 reports the regression results. We find that the coefficients of $Ind\_Area^*IV\_2012$ are 0.4338 and 1.0824, which are both significant at 1%. The results indicate that the ripple

**Table 6. The effect of CEO awards' gold content and social awareness of competitors.**

| Variables | R&D_1 | | R&D_2 | |
|---|---|---|---|---|
| | (1) | (2) | (3) | (4) |
| Ind_Area | 0.0751 (0.0524) | 0.4003*** (0.0301) | 0.0271 (0.1023) | 0.8275*** (0.0587) |
| Ind_Area*IV_2012 | 0.4338*** (0.0616) | | 1.0824*** (0.1202) | |
| Ind_Area*media | | -0.1463** (0.0696) | | -0.2796** (0.1360) |
| media | | 0.0985*** (0.0162) | | 0.1417*** (0.0317) |
| size | -0.0382*** (0.0064) | -0.0424*** (0.0064) | -0.1524*** (0.0125) | -0.1588*** (0.0126) |
| lev | -0.4825*** (0.0473) | -0.4882*** (0.0473) | -2.5281*** (0.0923) | -2.5357*** (0.0925) |
| roa | 2.5586*** (0.1647) | 2.5376*** (0.1648) | 1.2111*** (0.3216) | 1.1770*** (0.3221) |
| growth | 0.0131* (0.0079) | 0.0133* (0.0079) | 0.0833*** (0.0154) | 0.0840*** (0.0155) |
| cash | 0.4314*** (0.0508) | 0.4304*** (0.0508) | 0.7864*** (0.0992) | 0.7848*** (0.0993) |
| fcf | 0.0418*** (0.0121) | 0.0401*** (0.0121) | -0.0587** (0.0236) | -0.0607** (0.0237) |
| top5 | -0.0014** (0.0006) | -0.0016*** (0.0006) | -0.0039*** (0.0011) | -0.0041*** (0.0012) |
| age | -0.4457*** (0.0262) | -0.4428*** (0.0262) | -1.1657*** (0.0512) | -1.1611*** (0.0513) |
| soe | 0.0671*** (0.0238) | 0.0761*** (0.0238) | 0.0800* (0.0465) | 0.0990** (0.0446) |
| MRK | 0.0933*** (0.0053) | 0.0932*** (0.0053) | 0.1235*** (0.0103) | 0.1241*** (0.0104) |
| Industry effect | Y | Y | Y | Y |
| Time effect | Y | Y | Y | Y |
| N | 22932 | 22932 | 22932 | 22932 |
| Adj.R$^2$ | 0.3661 | 0.3658 | 0.4153 | 0.4137 |

Note: $IV\_2012$ is controlled by the time effect.

effects of CEO awards are more significant after 2012. If the social attention of award-winning CEOs' competitors is high, they may lack an incentive to undertake actions to improve performance to gain social recognition, which may weaken the ripple effects of CEO awards. Accordingly, we select the number of media reports on CEOs to measure social attention. The results show that the coefficient of $Ind\_Area*media$ is significantly negative, which confirms the results of theoretical analysis: when award-winning CEOs' competitors have higher social attention, the upward comparison motivation is weaker, which reduces the ripple effects of CEO awards.

## The impact of the similarity between award-winning CEOs and competitors

This study further discusses the similarities between award-winning CEOs and competitors from the perspectives of CEO age and the nature of enterprise property rights [22, 30–32]. If the cross-term between the dummy variable for CEO age and CEO awards is significantly negative, then the similarity between award-winning CEOs and competitors is an important factor affecting the ripple effects of CEO awards. When the competitor CEO is older than the award-winning CEO, the dummy variable for CEO age equals 1; otherwise, it equals 0.

The award-winning CEOs in this paper are from non-state-owned enterprises, and state-owned enterprises and non-state-owned enterprises have different management models and business objectives. If the cross term between the nature of property rights and CEO awards is significantly negative, it further shows that the similarity between award-winning CEOs and competitors has an impact on the ripple effects of CEO awards. When firms are state-owned enterprises, the dummy variable for the nature of property rights equals 1; otherwise, it equals 0.

$$R\&D_{i,t} = \alpha_0 + \alpha_1 Ind\_Area_{i,t} + \alpha_2 Ind\_Area_{i,t} * ceoage_{i,t}/soe_{i,t}$$
$$+ \sum_j \alpha_j X_{j,i,t} + \sum ind + \sum year + \varepsilon_{i,t} \tag{4}$$

Table 7 presents the regression results for Eq (4). The coefficients of cross terms $Ind\_Area*ceoage$ and $Ind\_Area*soe$ are -0.1499 and -0.2911, respectively, and both are significant at 1%. The results support H3 of this study, which means that the similarity of the comparison group will amplify the ripple effects of CEO awards.

## Industry competition and the effect of CEO awards

In industries with fierce competition, enterprises are more willing to compare themselves with other enterprises in the industry to address it [34, 35]. If the degree of industry competition has a positive moderating effect on the intraindustry ripple effects of CEO awards, industry competition is an important factor affecting the size of the ripple effects. This paper uses the Herfindahl-Hirschman index to measure industry competition. If the Herfindahl-Hirschman index is larger than the average level of other industries, $HHI$ equals 1, indicating that the degree of industry competition is high. $SO$ represents the business similarity between enterprises; it is 1 if it is larger than the average level of other industries and thus indicates a greater degree of competition in the industry. If the coefficient of $Ind\_Area*HHI$ or $Ind\_Area*SO$ is significantly positive, then the degree of industry competition has a positive moderating effect on the ripple effects of CEO awards.

$$R\&D_{i,t} = \alpha_0 + \alpha_1 Ind\_Area_{i,t} + \alpha_2 Ind\_Area_{i,t} * HHI_{i,t}/SO_{i,t}$$
$$+ \sum_j \alpha_j X_{j,i,t} + \sum ind + \sum year + \varepsilon_{i,t} \tag{5}$$

**Table 7. Similarity between the comparison group and the real effects of CEO awards.**

| Variables | R&D_1 | | R&D_2 | |
|---|---|---|---|---|
| | (1) | (2) | (3) | (4) |
| Ind_Area | 0.4216*** (0.0316) | 0.4311*** (0.0315) | 0.8907*** (0.0618) | 0.9235*** (0.0616) |
| Ind_Area*ceo_age | -0.2076*** (0.0674) | | -0.5119*** (0.1316) | |
| Ind_Area*soe | | -0.2911*** (0.0744) | | -0.7755*** (0.1454) |
| size | -0.0383*** (0.0064) | -0.0382*** (0.0064) | -0.1530*** (0.0125) | -0.1524*** (0.0125) |
| lev | -0.4841*** (0.0473) | -0.4830*** (0.0473) | -2.5319*** (0.0925) | -2.5293*** (0.0925) |
| roa | 2.5522*** (0.1648) | 2.5507*** (0.1648) | 1.1950*** (0.3221) | 1.1910*** (0.3222) |
| growth | 0.0135* (0.0079) | 0.0138* (0.0079) | 0.0841*** (0.0154) | 0.0849*** (0.0154) |
| cash | 0.4330*** (0.0508) | 0.4323*** (0.0508) | 0.7903*** (0.0993) | 0.7887*** (0.0993) |
| fcf | 0.0429*** (0.0121) | 0.0427*** (0.0121) | -0.0559*** (0.0236) | -0.0564** (0.0236) |
| top5 | -0.0015** (0.0006) | -0.0014** (0.0006) | -0.0039*** (0.0012) | -0.0040*** (0.0012) |
| age | -0.4416*** (0.0262) | -0.4425*** (0.0262) | -1.1558*** (0.0513) | -1.1572*** (0.0513) |
| soe | 0.0727*** (0.0238) | 0.0973*** (0.0246) | 0.0941** (0.0466) | 0.1594*** (0.0482) |
| MRK | 0.0940*** (0.0053) | 0.0940*** (0.0053) | 0.1252*** (0.0104) | 0.1253*** (0.0104) |
| Industry effect | Y | Y | Y | Y |
| Time effect | Y | Y | Y | Y |
| N | 22932 | 22932 | 22932 | 22932 |
| Adj.R$^2$ | 0.3650 | 0.3652 | 0.4134 | 0.4139 |

Note: ceo_age is a dummy variable that equals 1 if the CEO is older than the award-winning CEO and 0 otherwise.

Table 8 reports the regression results of Eq (5). The coefficients of the interaction terms Ind_Area*HHI and Ind_Area*SO are 0.4699 and 0.2298, respectively, and both are significant at 1%. This shows that when enterprises face strong industry competition, the ripple effects of

**Table 8. Industry competition and the effect of CEO awards.**

| Variables | R&D_1 | | R&D_2 | |
|---|---|---|---|---|
| | (1) | (2) | (3) | (4) |
| Ind_Area | -0.0636 (0.0560) | 0.2655*** (0.0372) | -0.1289 (0.1090) | 0.6517*** (0.0729) |
| Ind_Area*HHI | 0.4699*** (0.0643) | | 0.9501*** (0.1252) | |
| HHI | 0.3686*** (0.0194) | | 0.8334*** (0.0378) | |
| Ind_Area*SO | | 0.2298*** (0.0560) | | 0.2720** (0.01097) |
| SO | | 0.2078*** (0.0195) | | 0.2932*** (0.0383) |
| size | -0.0328*** (0.0064) | -0.0376*** (0.0064) | -0.1407*** (0.0125) | -0.1522*** (0.0125) |
| lev | -0.4667*** (0.0468) | -0.4741*** (0.0472) | -2.4930*** (0.0912) | -2.5170*** (0.0923) |
| roa | 2.5101*** (0.1631) | 2.4862*** (0.1644) | 1.0994*** (0.3175) | 1.1029*** (0.3218) |
| growth | 0.0112 (0.0078) | 0.0137* (0.0079) | 0.0792*** (0.0152) | 0.0847*** (0.0155) |
| cash | 0.4259*** (0.0503) | 0.4339*** (0.0507) | 0.7733*** (0.0979) | 0.7899*** (0.0992) |
| fcf | 0.0467*** (0.0120) | 0.0422*** (0.0121) | -0.0479** (0.0233) | -0.0578** (0.0236) |
| top5 | -0.0009 (0.0006) | -0.0014** (0.0006) | -0.0027** (0.0011) | -0.0039*** (0.0012) |
| age | -0.4398*** (0.0259) | -0.4521*** (0.0262) | -1.1533*** (0.0506) | -1.1740*** (0.0512) |
| soe | 0.0698*** (0.0236) | 0.0632*** (0.0238) | 0.0879* (0.0459) | 0.0808** (0.0466) |
| MRK | 0.0945*** (0.0052) | 0.0943*** (0.0053) | 0.1265*** (0.0102) | 0.1248*** (0.0104) |
| Industry effect | Y | Y | Y | Y |
| Time effect | Y | Y | Y | Y |
| N | 22932 | 22932 | 22932 | 22932 |
| Adj.R$^2$ | 0.3790 | 0.3698 | 0.3698 | 0.4154 |

CEO awards are more significant. In columns (3) and (4), the results still support the above conclusions. The above results confirm Hypothesis H4 of this paper: external pressure, represented by the degree of competition in the industry, will significantly amplify the ripple effects of CEO awards.

## Complementary and robustness tests

### Do the ripple effects of CEO awards increase firm value?

From the above conclusions, we find that CEO awards have significant ripple effects, that is, they lead to an increase in the R&D investment of award-winning CEOs' competitors. Now, using the Sobel intermediary factor test, we discuss whether the ripple effects of CEO awards can improve corporate performance and value. This method is constructed in a three-step equation [35]. The first step is regression Eq (2), and the second and third steps are as follows:

$$ROA_{i,t+1}/Tobin_{i,t+1} = \alpha_0 + \alpha_1 R\&D_{i,t} + \sum_j \alpha_j X_{j,i,t} + \sum ind + \sum year + \varepsilon_{i,t} \qquad (6)$$

$$ROA_{i,t+1}/Tobin_{i,t+1} = \alpha_0 + \alpha_1' Ind_{i,t}/IndArea_{i,t} + \sum_j \alpha_j X_{j,i,t} + \sum ind + \sum year + \varepsilon_{i,t} \qquad (7)$$

where *ROA* and *Tobin* are return on total assets and Tobin's Q, respectively; Eq (6) represents the second step to test whether R&D investment (mediating variable) improves firm performance and value; Eq (7) examines whether CEO awards (original variables) in the industry can improve firm performance and value. When the mediating effect test meets the following conditions, the intraindustry ripple effects of CEO awards can improve firm performance and value: (1) CEO awards increase the R&D investment of award-winning CEOs' competitors; (2) R&D investment can increase firm performance and value; and (3) the Sobel Z value is statistically significant. Table 9 reports the test results. We find that R&D investment significantly improves firm performance and value. The results show that the intraindustry ripple effects of CEO awards, namely, increasing the R&D investment by award-winning CEOs' competitors, can significantly improve firm performance and value.

### Robustness tests

**Number of patent applications as the explained variable.** This study further uses the number of patent applications of firms as an indicator of R&D output and tests the ripple effects of CEO awards. Then, we divide patent applications into invention patents and noninvention patents. Table 10 reports the test results. The total number of patent applications is used as the dependent variable in column (1), and the coefficient of *Ind_Area* is 0.1401, which is significant at the 1% level. Thus, the number of total patent applications of the treatment group increases by approximately 1.4 compared with enterprises without award-winning CEOs. In columns (2) and (3), the number of invention patents and noninvention patents are used as dependent variables, and the coefficients of *Ind_Area* are 0.0595 and 0.0805, respectively, indicating that compared with enterprises without award-winning CEOs in the same industry and region, the number of invention patents increases by 0.6 on average, 28.24% higher than the average. Additionally, noninvention patent applications increase by an average of 0.8, 11.25% higher than the average. This result further confirms that CEO awards have intraindustry ripple effects.

**PSM-DID test.** The likelihood of awards being given to CEOs in the industry is related to the characteristics of the industry. Although the above analysis controls for industry effects to

**Table 9. Do the ripple effects of CEO awards increase firm value?.**

| Variable | ROA | | | | Tobin's Q | | | |
|---|---|---|---|---|---|---|---|---|
| | (1) | (2) | (3) | (4) | (5) | (6) | (7) | (8) |
| R&D_1 | 0.0041*** (0.0003) | | | | 0.0141** (0.0055) | | | |
| R&D_2 | | 0.0005*** (0.0001) | | | | 0.0086*** (0.0028) | | |
| Ind | | | 0.0040*** (0.0007) | | | | 0.0292* (0.0155) | |
| Ind_Area | | | | 0.0057*** (0.0012) | | | | 0.0732*** (0.0243) |
| size | 0.0016*** (0.0003) | 0.0015*** (0.0003) | 0.0015*** (0.0003) | 0.0015*** (0.0003) | -0.3251*** (0.0054) | -0.3243*** (0.0054) | -0.3195*** (0.0054) | -0.3197*** (0.0054) |
| lev | -0.0955*** (0.0018) | -0.0970*** (0.0018) | -0.0982*** (0.0018) | -0.0983*** (0.0018) | 0.0553 (0.0401) | 0.0704* (0.0406) | -0.0164 (0.0375) | -0.0164 (0.0375) |
| roa | 0.0036 (0.0050) | 0.0061 (0.0050) | 0.0062 (0.0050) | 0.0076 (0.0050) | 1.3669*** (0.1043) | 1.3628*** (0.1046) | 1.3700*** (0.1043) | 1.3782*** (0.1041) |
| growth | 0.0019*** (0.0003) | 0.0019*** (0.0003) | 0.0019*** (0.0003) | 0.0019*** (0.0003) | 0.0317*** (0.0066) | 0.0311*** (0.0066) | 0.0339*** (0.0066) | 0.0338*** (0.0066) |
| cash | -0.0146*** (0.0020) | -0.0134*** (0.0020) | -0.0131*** (0.0020) | -0.0130*** (0.0020) | -0.1641*** (0.0431) | -0.1647*** (0.0431) | -0.1831*** (0.0429) | -0.1826*** (0.0429) |
| fcf | 0.0142*** (0.0005) | 0.0145*** (0.0005) | 0.0145*** (0.0005) | 0.0144*** (0.0005) | 0.0273*** (0.0101) | 0.0284*** (0.0101) | 0.0369*** (0.0099) | 0.0368*** (0.0099) |
| top5 | -0.0006*** (0.0000) | -0.0006*** (0.0000) | 0.0006*** (0.0000) | 0.0005*** (0.0000) | -0.0089*** (0.0005) | -0.0089*** (0.0005) | -0.0082** (0.0005) | -0.0082*** (0.0005) |
| age | -0.0026** (0.0010) | -0.0039*** (0.0011) | -0.0045*** (0.0010) | -0.0044*** (0.0010) | 0.3408*** (0.0224) | 0.3446*** (0.0225) | 0.3230*** (0.0222) | 0.3246*** (0.0222) |
| soe | -0.0028*** (0.0009) | -0.0026*** (0.0009) | -0.0025*** (0.0009) | -0.0025*** (0.0009) | -0.0546*** (0.0199) | -0.0544*** (0.0199) | -0.0454*** (0.0199) | -0.0446*** (0.0199) |
| MRK | 0.0012*** (0.0004) | 0.0016*** (0.0002) | 0.0017*** (0.0002) | 0.0015*** (0.0002) | -0.0527*** (0.0044) | -0.0526 (0.0044) | -0.0451*** (0.0043) | -0.0476*** (0.0044) |
| Industry effect | Y | Y | Y | Y | Y | Y | Y | Y |
| Time effect | Y | Y | Y | Y | Y | Y | Y | Y |
| N | 22932 | 22932 | 22932 | 22932 | 22932 | 22932 | 22932 | 22932 |
| Adj.R$^2$ | 0.2350 | 0.2272 | 0.2276 | 0.2274 | 0.3071 | 0.3072 | 0.2918 | 0.2922 |
| Sobel Z(1) | | | 11.4486*** | 10.3320*** | | | 2.5257** | 2.5087** |
| Sobel Z(2) | | | 3.9404*** | 3.8734*** | | | 3.0169*** | 2.9865*** |

Note: If using *R&D_1* as the intermediary variable, it is Sobel Z(1); if using *R&D_2* as the intermediary variable, it is Sobel Z(2). ROA and Tobin's Q are the next period values, and the control variable is the last period value.

alleviate this problem, to further ensure the robustness of the conclusions, we use the propensity score matching (PSM) method to perform one-to-one matching between the subsamples with (experimental group) and without (control group) award-winning CEOs. Then, we use DID to eliminate the endogeneity problem. The matching equation includes not only the control variables mentioned above but also the number of analysts tracking and media coverage. We choose the two years before an award is given as the preaward period and the two years after as the postaward period. The experimental group and the control group included 253 firms, and the total period was 4 years. The detailed PSM-DID results are shown in Table 11, and the coefficients of *time*time**treat* are significantly positive at the 5% level, indicating that the conclusions of this paper remain valid after alleviating the endogeneity problem.

**Table 10. Number of patent applications as the explained variable.**

| Variable | Total | Invention | Noninvention |
|---|---|---|---|
| | (1) | (2) | (3) |
| Ind_Area | 0.1401*** (0.0454) | 0.0595*** (0.0106) | 0.0805** (0.0390) |
| size | 0.2299*** (0.0100) | 0.0528*** (0.0023) | 0.1770*** (0.0086) |
| lev | 0.3614*** (0.0741) | 0.0227 (0.0174) | 0.3388*** (0.0636) |
| roa | 1.6467*** (0.2582) | 0.2675*** (0.0605) | 1.3792*** (0.2216) |
| growth | 0.0021 (0.0124) | 0.0056* (0.0029) | -0.0035 (0.0010) |
| cash | 0.1468* (0.0796) | -0.0188*** (0.0186) | 0.1657** (0.0683) |
| fcf | 0.0961*** (0.0189) | 0.0118*** (0.0044) | 0.0842*** (0.0023) |
| top5 | 0.0043*** (0.0009) | -0.0001 (0.0002) | 0.0044*** (0.0008) |
| age | -0.2499*** (0.0411) | -0.0481*** (0.0096) | -0.2017*** (0.0353) |
| soe | 0.0286 (0.0373) | 0.0342*** (0.0087) | 0.0044 (0.0321) |
| MRK | 0.0651*** (0.0083) | 0.0188*** (0.0019) | 0.0463*** (0.0071) |
| Industry effect | Y | Y | Y |
| Time effect | Y | Y | Y |
| N | 22932 | 22932 | 22932 |
| Adj.R$^2$ | 0.1222 | 0.1244 | 0.1027 |

Note: To make the results more intuitive, the number of patents is divided by 10.

**Other robustness tests.** There is a large variability in R&D investment among different enterprises, and the overall distribution is right skewed. To reduce the interference of this biased distribution on the results, this paper uses a negative binomial distribution regression to re-estimate the impact of CEO awards on R&D investment. The regression results are reported in columns (1) and (2) of Table 12. We find that the research conclusions of this paper remain valid after considering the right-sided distribution of R&D investment. Further, this paper sets four dummy variables and examines the changes in the R&D investment of enterprises before and after CEOs in the industry receives awards. Ind_Area[−2] indicates that at least one CEO in the industry is given an award for the following two years, and so on, Ind_Area[+2] indicating that there is a CEO award in the industry for the previous two years. If a CEO award indeed motivates other enterprises in the industry, the coefficients of Ind_Area[−1] and Ind_Area[−2] should not be significant or negative, while the coefficients of Ind_Area[+1] and Ind_Area[+2] should be significantly positive. The results in Table 12 show

**Table 11. PSM-DID.**

| Variable | R&D_1 | R&D_2 |
|---|---|---|
| | (1) | (2) |
| time*treat | 0.1701** (0.0737) | 0.2706** (0.1226) |
| Control Variable | Y | Y |
| Industry effect | Y | Y |
| Time effect | Y | Y |
| Individual effect | Y | Y |
| N | 1856 | 1856 |
| Adj.R$^2$ | 0.1006 | 0.0945 |

Note: time is controlled by the time effect; treat is controlled by the individual effect.

**Table 12. Other robustness tests.**

| Variable | Negative binomial regression | | Dynamic test | |
|---|---|---|---|---|
| | R&D_1 | R&D_2 | R&D_1 | R&D_2 |
| | (1) | (2) | (3) | (4) |
| Ind_Area | 0.1829*** (0.0276) | 0.1848*** (0.0303) | | |
| Ind_Area[−2] | | | 0.0036 (0.0551) | -0.0574 (0.1046) |
| Ind_Area[−1] | | | 0.0809 (0.0501) | 0.0964 (0.0950) |
| Ind_Area[+1] | | | 0.1296** (0.0448) | 0.2188** (0.0851) |
| Ind_Area[+2] | | | 0.1278*** (0.0446) | 0.3099*** (0.0846) |
| Control Variable | Y | Y | Y | Y |
| Industry effect | Y | Y | Y | Y |
| Time effect | Y | Y | Y | Y |
| N | 22932 | 22932 | 22932 | 22932 |
| Pseudo $R^2$/Adj.$R^2$ | 0.2072 | 0.1812 | 0.3586 | 0.4175 |

that the R&D investment of competitors increased significantly after CEOs were given awards, which further confirms the conclusions of this paper.

**Excluding an alternative hypothesis.** Shi et al. [22] found that CEO awards will encourage competitors to undertake acquisition activities more frequently to increase their social recognition. To exclude this alternative hypothesis, this paper uses whether companies launched acquisition programs as the dependent variable. The results show that CEO awards in China do not lead to more acquisition activities, indicating that CEO awards do not encourage other CEOs in the industry to increase their social recognition through more radical mergers and acquisitions. Deng Bofu et al. [23] found that in China, awards for firm performance reduce the earnings management of award-winning enterprises while increasing the earnings management of other enterprises in the industry. Li et al. [36] also find that CEOs are more likely to engage in financial misconduct after the media names them as being among the best business leaders. But, Hou [37] find that corporate social responsibility (CSR) awards stimulate firm to achieve financial results superior to those of firms which do not pursue CSR awards. This study examines whether CEO awards will affect the earnings management of other enterprises in the industry. The empirical results in Table 13 show that there are some differences between CEO awards and corporate awards, and CEO personal awards do not lead to a higher earnings management level.

**Table 13. Excluding an alternative hypothesis.**

| Variable | M&A | | Earnings Management | |
|---|---|---|---|---|
| | (1) | (2) | (3) | (4) |
| Ind | 0.0301 (0.0202) | | -0.0021* (0.0011) | |
| Ind_Area | | 0.0518 (0.0317) | | 0.0003 (0.0017) |
| Control Variable | Y | Y | Y | Y |
| Industry effect | Y | Y | Y | Y |
| Time effect | Y | Y | Y | Y |
| N | 22932 | 22932 | 22932 | 22932 |
| Pseudo $R^2$/Adj.$R^2$ | 0.0182 | 0.0182 | 0.1043 | 0.1041 |

Note: M&A is a dummy variable that equals 1 when the enterprise implemented an acquisition program and 0 otherwise. Earnings management is calculated according to the modified Jones model [38].

## Conclusion and implications

Considering the social comparison theory and the social background of traditional Chinese culture, this study analyzes the intraindustry ripple effects of CEO awards from the perspective of enterprise R&D investment. Unlike the findings of Shi et al [22], we find that CEO awards are an incentive in China because of the country's cultural background; they can significantly increase the R&D investment of other enterprises in the industry. Based on social comparison theory, this paper further finds that the "gold content" of CEO awards, the social attention of award-winning CEOs' competitors, the similarity between award-winning CEOs and competitors, the property rights of enterprises, and external comparative pressure are important factors affecting the size of ripple effects. Specifically, "gold content", similarity and industrial competition significantly enlarge the intraindustry ripple effects of CEO awards. The social attention of award-winning CEOs' competitors weakens the intraindustry ripple effects of CEO awards. This paper confirms these ripple effects by examining the number of patent applications. The empirical evidence also finds that the ripple effects of CEO awards significantly improve firm performance and value.

The theoretical insight is that the existing literature pays a great deal of attention to the internal impact of CEO awards on their enterprises, including individual effects and organizational effects, and less attention to the ripple effects of CEO awards. This paper considers social comparison theory alongside corporate finance-related issues and finds that CEO awards have obvious intraindustry ripple effects, which provides a new perspective for understanding the impact of CEO awards. This paper further discusses the factors affecting the size of the ripple effects and finds that the "gold content" of CEO awards, the social attention of other CEOs, the similarity between comparison groups and the degree of industry competition are important factors affecting the ripple effects of CEO awards; these findings provide a reference for better exploiting the ripple effects of CEO awards. According to Maslow's hierarchy of needs, CEOs need respect and self-realization after their basic physiological and security needs are met. The findings of this paper show that in the traditional Chinese cultural background, CEO awards have positive ripple effects on other enterprises. Therefore, firms should pay attention to the traditional Chinese culture and social background when implementing management incentive arrangements.

## Limitations

Due to data limitation, we do not analysis the impacts of different type of CEO awards, such as the different impacts of award provide by government and social media. Caiffa et al. [39] point out that CEOs' celebrity not only related with media content, but also some other contents. The intraindustry ripple effects maybe more significant in state-owned enterprises if the awards provide by government, and the political relation between government and state-owned enterprises may decrease the ripple effects of government's awards. Then, we do not consider the impacts of leader humility, existing study shows that leader humility significantly impacts on subordinates [40]. In future, we will further discuss how leader humility impacts on the ripple effects of CEO awards.

## Supporting information

**S1 File. Data and code to reach our conclusions.**
(RAR)

## Author Contributions

**Conceptualization:** Yu Wu.

**Data curation:** Yu Wu.

**Formal analysis:** Yu Wu.

**Supervision:** Yingyi Hu.

**Writing – original draft:** Yu Wu.

**Writing – review & editing:** Yu Wu.

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
