## [Decision Letter · Decision Letter 0]

6 May 2021

PONE-D-21-09184

Chinese-style incentives: The intraindustry ripple effects of CEO awards

PLOS ONE

Dear Dr. Wu,

Thank you for submitting your manuscript to PLOS ONE. After careful consideration, we feel that it has merit but does not fully meet PLOS ONE’s publication criteria as it currently stands. Therefore, we invite you to submit a revised version of the manuscript that addresses the points raised during the review process.

We look forward to receiving your revised manuscript.

Kind regards,

Bing Xue, Ph.D.

Academic Editor

PLOS ONE

Journal Requirements:

We suggest you thoroughly copyedit your manuscript for language usage, spelling, and grammar. If you do not know anyone who can help you do this, you may wish to consider employing a professional scientific editing service. 

We note you have included a table to which you do not refer in the text of your manuscript. Please ensure that you refer to Table 5 in your text; if accepted, production will need this reference to link the reader to the Table.

Thank you for stating the following financial disclosure:

In your Data Availability statement, you have not specified where the minimal data set underlying the results described in your manuscript can be found. PLOS defines a study's minimal data set as the underlying data used to reach the conclusions drawn in the manuscript and any additional data required to replicate the reported study findings in their entirety. All PLOS journals require that the minimal data set be made fully available. For more information about our data policy, please see http://journals.plos.org/plosone/s/data-availability.

Reviewers' comments:

Reviewer's Responses to Questions

**Comments to the Author**

1. Is the manuscript technically sound, and do the data support the conclusions?

Reviewer #1: Yes

Reviewer #2: Yes

Reviewer #3: Yes

Reviewer #4: Yes

Reviewer #5: Yes

2. Has the statistical analysis been performed appropriately and rigorously? 

Reviewer #1: Yes

Reviewer #2: Yes

Reviewer #3: Yes

Reviewer #4: Yes

Reviewer #5: Yes

3. Have the authors made all data underlying the findings in their manuscript fully available?

Reviewer #1: Yes

Reviewer #2: Yes

Reviewer #3: Yes

Reviewer #4: Yes

Reviewer #5: No

4. Is the manuscript presented in an intelligible fashion and written in standard English?

Reviewer #1: Yes

Reviewer #2: Yes

Reviewer #3: Yes

Reviewer #4: Yes

Reviewer #5: Yes

5. Review Comments to the Author

Reviewer #1: Great work. It makes a very insightful reading . The analysis are very well done and the recommendation is apt. A good demonstration of the principles of research. Looking forward for more relate and extend the work

Reviewer #2: The paper is well written and novel, making a good contribution. To make the paper easier to comprehend I suggest the authors shorten the introduction Literature as it is too long.Improves these parts

Reviewer #3: I found that this paper is very interesting about The Intra industry ripple effects of CEO awards. This study examines the Intra-industry ripple effects of CEO awards from the viewpoint of

enterprise R&D investment. That one is very important regarding R&D investment and it significantly improves firm performance and value. This article is well written.

Reviewer #4: I found this study interesting. There are few changes recommended in the manuscript.

1. The study is based on the majority of very old articles, it would be good to include somelatest researches in the field.

2. It is recommended that a native English speaker may conduct minor revisions.

3. The manuscript should include limitations of the study.

Reviewer #5: Thank you for providing me an opportunity to read an interesting article. The article has potential and its contributions are strong. however, there is a small concern that I have the literature cited in the article is a little old. There is only one study from 2020 that is discussed. I suggest improving the literature quality. Good Luck

6. PLOS authors have the option to publish the peer review history of their article (what does this mean?). If published, this will include your full peer review and any attached files.

Reviewer #1: **Yes: **Haitham Medhat Aboulilah

Reviewer #2: No

Reviewer #3: No

Reviewer #4: No

Reviewer #5: **Yes: **Shahid Ali

---

## [Author Response · Author response to Decision Letter 0]

20 May 2021

Dear Academic Editor and Reviewers:

Thank you for your letter and the reviewers’ comments on our manuscript entitled "Chinese-style incentives: The intraindustry ripple effects of CEO awards" (ID: PONE-D-21-09184). Those comments are very helpful for revising and improving our paper, as well as the important guiding significance to other research. We have studied the comments carefully and made corrections which we hope meet with approval. The main corrections are in the manuscript and the responds to the journal requirements and the reviewers’ comments are as follows (the replies are highlighted in blue).

Response to Journal Requirements

Q1: Please ensure that your manuscript meets PLOS ONE's style requirements, including those for file naming. 

Response: We modify our manuscript according to PLOS ONE's style guidelines and articles already published by PLOS ONE.

Q2: We suggest you thoroughly copyedit your manuscript for language usage, spelling, and grammar.

Response: Our paper has been copyedited by AJE before we submit to PLOS ONE (Order ID: JPHN7CYT). And, we carefully check again to make sure the precise of language usage, spelling, and grammar of our manuscript.

Q3: We note you have included a table to which you do not refer in the text of your manuscript. Please ensure that you refer to Table 5 in your text; if accepted, production will need this reference to link the reader to the Table.

Response: Thanks editor for point out our mistake. We carefully check table in our manuscript to ensure the accuracy of table number.

Q4: Statement on funding

Response: The authors received no specific funding for this work. And, we do our statements in new cover letter.

Q5: Data availability 

Response: We upload data set as Supporting Information files, named “Supplementary file -Data”.

Q6: Please review your reference list to ensure that it is complete and correct. 

Response: We carefully check our references again to make sure they are complete and correct. And, we add DOI to each reference, except 4 important Chinese literatures. And, we add the website where can download those Chinese literatures since there are no DOI for them.

Response to Reviews

Reviewer #1: Great work. It makes a very insightful reading. The analysis are very well done and the recommendation is apt. A good demonstration of the principles of research. Looking forward for more relate and extend the work.

Response: We are very glad that you like our paper, and we believe this topic are very important both for economic and social studies. And, we are trying our best to do further study on this topic.

Reviewer #2: The paper is well written and novel, making a good contribution. To make the paper easier to comprehend I suggest the authors shorten the introduction Literature as it is too long. Improves these parts.

Response: Thanks for your insightful advice. In introduction literature, we delete some unimportant parts. 

Reviewer #3: I found that this paper is very interesting about The Intra industry ripple effects of CEO awards. This study examines the Intra-industry ripple effects of CEO awards from the viewpoint of enterprise R&D investment. That one is very important regarding R&D investment and it significantly improves firm performance and value. This article is well written.

Response: Thanks for your high compliment. We are trying our best to make it better. 

Reviewer #4: I found this study interesting. There are few changes recommended in the manuscript.

1. The study is based on the majority of very old articles, it would be good to include some latest researches in the field.

2. It is recommended that a native English speaker may conduct minor revisions.

3. The manuscript should include limitations of the study.

Response: Thanks for your helpful recommendations. Our paper is combine the social comparison theory and corporate finance, this is a quite new topic to discuss the effects of nonfinancial incentives, which means the related literatures are less. We try our best to find more related and new literatures to prove our theoretical base and conclusions. We find 4 related literature are published in recent:

1. Tony C-T H.The relationship between corporate social responsibility and sustainable financial performance: firm‐level evidence from Taiwan. Corporate Social Responsibility and Environmental Management. 2019; 26(1):19-28. doi:10.1002/csr.1647

2. Jiangyan L, Wei S, Brian C, Xiwei Y, Xin Q. CEO awards and Financial Misconduct. Journal of Management. Advance online publication 2020. doi:10.1177/2F0149206320921438

3. Caiffa M, Farina V, Fattobene L. All that glitters is not gold: CEOs' celebrity beyond media content. International Journal of Finance & Economics. 2019; 25(3): 444-460. doi:10.1002/ijfe.1761

4. Xin Q, Chen C, Yam KC, Mingpeng H, Dong J. The double-edged sword of leader humility: Investigating when and why leader humility promotes versus inhibits subordinate deviance. Journal of Applied Psychology. 2020;105(7):693-712. doi:10.1037/apl0000456 

We carefully check again to make sure the precise of language usage, spelling, and grammar of our manuscript. And, we find a professional scientific editing service agency (AJE) to conduct language revisions for our paper.

We add the limitations about our paper in resubmitted manuscript. The main contains are as follows:

 Due to data limitation, we do not analysis the impacts of different type of CEO awards, such as the different impacts of award provide by government and social media. Caiffa et al. [39] point out that CEOs' celebrity not only related with media content, but also some other contents. The intraindustry ripple effects maybe more significant in state-owned enterprises if the awards provide by government, and the political relation between government and state-owned enterprises may decrease the ripple effects of government’s awards. Then, we do not consider the impacts of leader humility, existing study shows that leader humility significantly impacts on subordinates. [40] In future, we will further discuss how leader humility impacts on the ripple effects of CEO awards.

Reviewer #5: Thank you for providing me an opportunity to read an interesting article. The article has potential and its contributions are strong. however, there is a small concern that I have the literature cited in the article is a little old. There is only one study from 2020 that is discussed. I suggest improving the literature quality. Good Luck

Response: Thanks for your high compliment. Our paper is combine the social comparison theory and corporate finance, this is a quite new topic to discuss the effects of nonfinancial incentives, which means the related literatures are less. We try our best to find more related and new literatures to prove our theoretical base and conclusions. We add some new literatures into paper, as follows:

1. Tony C-T H.The relationship between corporate social responsibility and sustainable financial performance: firm‐level evidence from Taiwan. Corporate Social Responsibility and Environmental Management. 2019; 26(1):19-28. doi:10.1002/csr.1647

2. Jiangyan L, Wei S, Brian C, Xiwei Y, Xin Q. CEO awards and Financial Misconduct. Journal of Management. Advance online publication 2020. doi:10.1177/2F0149206320921438

3. Caiffa M, Farina V, Fattobene L. All that glitters is not gold: CEOs' celebrity beyond media content. International Journal of Finance & Economics. 2019; 25(3): 444-460. doi:10.1002/ijfe.1761

4. Xin Q, Chen C, Yam KC, Mingpeng H, Dong J. The double-edged sword of leader humility: Investigating when and why leader humility promotes versus inhibits subordinate deviance. Journal of Applied Psychology. 2020;105(7):693-712. doi:10.1037/apl0000456 

Reference

39. Caiffa M, Farina V, Fattobene L. All that glitters is not gold: CEOs' celebrity beyond media content. International Journal of Finance & Economics. 2019; 25(3): 444-460. doi:10.1002/ijfe.1761

40. Xin Q, Chen C, Yam KC, Mingpeng H, Dong J. The double-edged sword of leader humility: Investigating when and why leader humility promotes versus inhibits subordinate deviance. Journal of Applied Psychology. 2020;105(7):693-712. doi:10.1037/apl0000456

---

## [Editor Report · Decision Letter 1]

25 May 2021

Chinese-style incentives: The intraindustry ripple effects of CEO awards

PONE-D-21-09184R1

Dear Dr. Wu,

We’re pleased to inform you that your manuscript has been judged scientifically suitable for publication and will be formally accepted for publication once it meets all outstanding technical requirements.

Kind regards,

Bing Xue, Ph.D.

Academic Editor

PLOS ONE
---

## [Editor Report · Acceptance letter]

27 May 2021

PONE-D-21-09184R1 

Chinese-style incentives: The intraindustry ripple effects of CEO awards 

Dear Dr. Wu:

I'm pleased to inform you that your manuscript has been deemed suitable for publication in PLOS ONE. Congratulations! Your manuscript is now with our production department. 

Kind regards, 

on behalf of

Professor Bing Xue 

Academic Editor

PLOS ONE